# The Role of Gut Microbiota in Heart Failure: When Friends Become Enemies

**DOI:** 10.3390/biomedicines10112712

**Published:** 2022-10-26

**Authors:** Rossella Cianci, Laura Franza, Raffaele Borriello, Danilo Pagliari, Antonio Gasbarrini, Giovanni Gambassi

**Affiliations:** 1Department of Translational Medicine and Surgery, Catholic University of Rome, Fondazione Policlinico Universitario A. Gemelli, IRCCS, 00168 Rome, Italy; 2Emergency Medicine Unit, Catholic University of Rome, Fondazione Policlinico Universitario A. Gemelli, IRCCS, 00168 Rome, Italy; 3Medical Officer of the Carabinieri Corps, Health Service of the Carabinieri General Headquarters, 00197 Rome, Italy

**Keywords:** heart failure, gut microbiota, innate immunity, adaptive immunity, immune modulation

## Abstract

Heart failure is a complex health issue, with important consequences on the overall wellbeing of patients. It can occur both in acute and chronic forms and, in the latter, the immune system appears to play an important role in the pathogenesis of the disease. In particular, in the forms with preserved ejection fraction or with only mildly reduced ejection fraction, some specific associations with chronic inflammatory diseases have been observed. Another interesting aspect that is worth considering is the role of microbiota modulation, in this context: given the importance of microbiota in the modulation of immune responses, it is possible that changes in its composition may somewhat influence the progression and even the pathogenesis of heart failure. In this narrative review, we aim to examine the relationship between immunity and heart failure, with a special focus on the role of microbiota in this pathological condition.

## 1. Introduction

Heart failure (HF) is a major health issue with severe consequences on morbidity and mortality all over the world [1]. In its acute form, HF is associated with inflammatory markers, but in chronic HF, an altered inflammatory status and some pro-inflammatory mediators are considered to play a pivotal role in the pathogenesis of this condition [2].

Overall, many conditions in which the immune system is dysregulated have been linked to the development of HF. These pathologies range from lupus erythematosus to diabetes and obesity [3,4,5]. Part of the changes in the function of the immune system may also not be directly determined by the inflammatory condition itself but through modifications of the gut microbiota, which is a key factor in the regulation and modulation of the immune system [6,7].

HF can be associated or not to a reduced ejection fraction (EF). According to the new 2021 ESC guidelines, HF with a reduced EF (HFrEF) is characterized by systolic dysfunction associated with an EF of less than or equal to 40% [8]. This form is associated with important changes not only in the cardiac morphology and function but also in the cardiac metabolism, particularly to an increased glycolysis [9].

Patients with HF with mildly reduced EF (HFmrEF) represent a new category identified by the new ESC guidelines and are defined as the patients with EF between 41% and 49% in concomitance with symptoms and/or signs of HF [8]. This group of patients have epidemiological features which are similar to those with HFrEF, even though they have a lower mortality [8].

HF with preserved EF (HFpEF) accounts for about 50% of all cases of HF and is likely to become even more prevalent in the next decades [10]. HFpEF is characterized by signs and symptoms of heart failure and a left ventricular ejection fraction (LVEF) greater than 50% [8].

The etiology of heart failure involves all the conditions that can induce an impairment of the heart’s left or right ventricular filling, blood ejection, or filling pressure [11]. These conditions include both diseases causing a direct myocardial injury or functional anomaly, such as acute or chronic ischemic cardiac disease, congenital or acquired cardiomyopathies, valve diseases, inflammatory heart diseases, infiltrative or accumulation disorders, cardiotoxicity due to chemotherapy or drug abuse or arrhythmias, and conditions which cause an indirect involvement of heart such as chronic pulmonary hypertension, anemia, endocrine and nutritional disorders, autoimmune diseases [8,11].

While HFrEF is characterized primarily by a direct myocardial injury and metabolic changes, HFpEF appears to be linked to conditions which determine chronic inflammation (e.g., obesity, atherosclerosis, dyslipidemia, metabolic syndrome, and diabetes) and patients who suffer from HFpEF present an increase in several systemic markers of inflammation, such as serum levels of reactive C-protein (CRP), or interleukin-6 (IL-6) [12].

The aim of the present narrative review is to analyze the role of the gut microbiota and immuno-inflammatory factors in HF. We will then discuss the role of the immune mediators in cardiovascular (CV) health in general and the possible mechanisms through which they may promote or inhibit the development of HF.

## 2. HF and Inflammation

Several inflammatory cells and mediators of inflammation have been involved in the pathogenesis of HF. A special role is played by the immune system that is involved in both innate and adaptive subsets: T regulatory (Tregs), T helper 17 (Th17) cells, dendritic cells (DCs), macrophages, neutrophils, and eosinophils. These cells are able to produce several cytokines and mediators of inflammation that, in turn, can worsen tissue injury and perpetuate the inflammation through the self-activation of the same immune cells. Among these mediators, a special role is played by interleukins (IL)-1β, IL-6, IL-10, IL-13, tumor necrosis factor-α (TNF-α), transforming growth (TGF)-β, matrix metalloproteinase (MMP)-2 and MMP-9, and Toll-like receptors (TLRs). Thus, the cardiac microenvironment during myocardial injury is characterized by an imbalance between pro-inflammatory and anti-inflammatory immune factors. In this scenario, each player can be a friend or a foe, according to the substances present at the injury scene. Inflammation can directly affect the heart’s structure, increasing its stiffness via IL-1β and IL-6, IL-13, TNF-α, intercellular adhesion molecule-1 (ICAM-1), vascular cell adhesion molecule-1 (VCAM-1), chemokine receptor type 2 (CCR2), and chemokine ligand 2 (CCL2), while decreasing the activity of the collagen degrading MMP. Inflammatory pathways also act indirectly, negatively affecting autophagic mechanisms in the heart [13]. Several anti- and pro-inflammatory cytokines are involved in the pathogenesis and progression of HF, affecting myocardial remodeling and myocyte functions.

Elevated levels of TNF-α were the first to be linked to HFrEF [14]. Furthermore, IL-1β, IL-6, IL-8, monocyte chemo-attractant peptide-1 (MCP-1), and macrophage inflammatory protein-1α (MIP-1α) are also elevated in HF patients and can promote an inflammatory status, recruiting pro-inflammatory cells [15]. On the contrary, IL-10 represents an anti-inflammatory cytokine, possibly down-regulating the production of pro-inflammatory mediators, inhibiting the reactive oxygen products and blocking the effects of TNF-α [16]. In general, in HF, the levels of both pro-inflammatory (IL-6, TNF-α) and anti-inflammatory (IL-10) cytokines are increased, perhaps through a feedback mechanism. The IL-10/TNF-α ratio is, however, lower in HF patients than in healthy subjects, perhaps due to an insufficient production of IL-10 in response to TNF-α levels, which is responsible for disease progression [17]. Yamahoka et al. have shown that by stimulating mononuclear cells in vitro with lipopolysaccharide (LPS), the levels of produced IL-10 were higher in HF patients than in healthy subjects [18]. The reported increase in IL-10 level in patients with HF is also associated with a high class of NYHA [19]. The production of TNF-α, instead, does not increase. It is unknown why in HF the IL-10 levels are increased [20], but it is possible that TNF-α, as well the catecholamines and other humoral factors, may trigger IL-10 production too [19]. Moreover, genetic factors, such as IL-10 gene promoter polymorphisms, can also influence IL-10 production [20]. 

Experimental studies in mice have revealed that, after a myocardial infarction, IL-10 improves left ventriculus (LV) contractility, reducing the infarcted zone [21]. Yamaoka et al. have showed that both circulating IL-10 and surface IL-10 receptor (IL-10R) on mononuclear leukocytes and TNF-α are increased in patients with HF [22]. Furthermore, in HF, the production of IL-10 by mononuclear leukocytes is significantly increased by LPS stimulation. However, data have not been confirmed and are also quite contrasting [23]. Gorzin et al., for example, recently demonstrated that IL-10 expression was lower in HF, in respect to healthy controls, and no relation between IL-10 and the severity and/or etiology of HF was found.

In HF, it has been demonstrated that free oxygen radicals increase with the advanced states of the disease [21] and are correlated with IL-6 levels, but not with IL-10 ones. IL-6 correlates with HF severity and its prognosis, with high class of NYHA and with the rate of hospital readmission [22]. Both IL-6 and IL-10 resulted reduced after HF treatment [22]. However, this is not shown for TNF-α. Overall, CV mortality is associated with increased pro-inflammatory cytokines levels [23].

Thus, myocardial inflammation following myocardial injury is associated with a strong modification of the cardiac microenvironment, in which several immune and inflammatory cells and mediators of inflammation are recruited and activated. Inflammation guides intense cellular tissue homing trafficking and the consequent production of pro-inflammatory cytokines. In this inflamed scenario, there is a mutual relationship between cells and their related cytokines present at the site of cardiac injury. Such a cytokine-cocktail is able to induce several T-cells to differentiate into pro-inflammatory subsets; T-cells’ plasticity is responsible for the induction of other pro-inflammatory pathways. Furthermore, to complicate this intricate relationship, each cell can work in multiple ways, depending on the environment in which it is placed: known ‘enemy’ cells may become friends and, after prolonged inflammatory status, ‘friend’ cells may become enemies.

A summary of the effects of pro-inflammatory and anti-inflammatory cytokines in HF is reported in Table 1.

## 3. Impact of Immunity on LV Function and Remodeling

Even though it is common to generally speak about HF, without taking into account the value of the EF, evidence supports the hypothesis that HFpEF and HFrEF have to be considered as different diseases. In this way, the European Cardiology Society (ESC) guidelines of 2016 classify HF in two main categories, HFrEF and HFpEF, with a so-called “grey area” between the two [24]. This mentioned “grey area” has been eliminated by the last guidelines of 2021, which have defined the category of HFmrEF [8].

While in HFrEF, therapy has been proven to reduce mortality and morbidity, the efficacy of therapy in HFpEF is still not ideal, even though diagnosis of this condition is becoming more accurate and data on the use of diuretics in this type of HF are encouraging [25]. Yet, HFrEF has a higher mortality rate than HFpEF and in the latter condition causes of death are mostly non cardiovascular [26]. Moreover, the etiologies of the diseases are very different: HFrEF is secondary to an injury or disease which affects the heart, determining a reduced ventricular contraction. Even though in about one third of the cases, the causes are non-CV (e.g., thyroid disorders, sarcoidosis), two thirds of all cases are consequence of CV disorders, mostly coronary disease [27]. HFpEF instead is a more complex disease, which probably should be considered as part of a syndrome involving the whole organism [28] and has even been sub-classified in three different phenotypes: the cardiorenal, the obesity-cardiometabolic, and the natriuretic peptide deficiency phenotype [29]. Even though there are many differences in the physiopathology of HFrEF and HFpEF, some of the risk factors in the development of both these disease are shared, such as hypertension, smoking, obesity, diabetes and metabolic syndrome, but while in HFpEF patients are mostly older females and usually present with a minor CV history (usually significant only for hypertension and atrial fibrillation), patients with a HFrEF are instead younger males with a history of left ventricular hypertrophy, bundle branch block, previous myocardial infarction, and smoking [27].

When studying, instead, immune differences in HFpEF and HFrEF data are still not conclusive, but it seems that there might be differences in the mechanisms underlying LV remodeling in these two subsets [30]. In HFrEF, cardiac remodeling is based primarily on a local insult (e.g., drug toxicity, ischemia, infections) which can directly activate the immune system at local tissues that may be considered as an ‘immune niche’, that, in turn, is able to interplay with several different types of cytokines, mediators of inflammation, and bacterial components. Instead, when a patient develops HFpEF, inflammation is generalized and does not directly affect the myocardiocytes [3]. An example of such a situation is brought to us by Tschöpe et al., which demonstrated how parvovirus B19 can determine HF also when not directly affecting myocardiocytes, but while the direct infection causes HFrEF, the forms of HF following a generalized infection which has not directly affected the heart are more often than not HFpEF [31]. Another proof of the different mechanisms underlying the differences in the remodeling process in HFpEF vs. HFrEF is the fact that diabetic patients who experience LV remodeling in the context of HF are the ones who present a preserved EF, while the ones who have a reduced EF do not experience as much remodeling [32].

Overall, LV remodeling is still not completely understood, and the immune niche of the heart plays an important role, particularly in HFpEF, while HFrEF seems to be more of a mechanical process [33]. The way the immune system acts in remodeling the LV seems to differ also based on the etiology underlying HF: indeed, ischemic forms are characterized by a stronger native response, while non-ischemic HF is characterized by a prevalent response of B and T-lymphocytes [34].

## 4. The Interplay between Immune Response and HF: Main Actors of Innate and Adaptive Immunity

Innate immunity may play a role in the pathogenesis of HF and macrophages are involved in both acute and chronic myocardial injury. Two distinct populations of macrophages are present, CCR2^−^ myocardial resident macrophages and CCR2^+^ macrophages, that are recruited from peripheral circulation [35]. In the setting of acute injury, resident macrophages are able to recognize damage-associated molecular patterns (DAMPs), including cellular components of the dying cardiomyocytes, and then produce specialized pro-resolving mediators (SPMs). SPMs induce the recruitment and activation of neutrophils that participate in tissue repair, and these mediators are also able to reprogram macrophages function to drive the resolution of inflammation. Moreover, macrophages produce TGF-β and IL-10 that activate Tregs and induce fibroblast migration, with the consequent collagen deposition [35]. In the setting of chronic inflammation, CCR2^+^ circulating macrophages contribute to the worsening of LV systolic dysfunction [36]. CCR2^+^ macrophages are also capable to produce TGF-β, which is responsible for the fibroblast-driven excess of myocardial collagen deposition, leading to the fibrotic scar, and other fibrotic agents such as galectin-3 and osteopontin [37,38]. Another subset of cells participating in innate immunity is represented by neutrophils, considered pro-inflammatory cells. Indeed, neutrophils seem to enhance ischemic damage and ischemia-reperfusion injury, releasing reactive oxygen species (ROS) [39], such as myeloperoxidase and proteases. Moreover, they are able to recruit monocytes and macrophages at the site of injured myocardial tissue. Yet, in experimental myocardial infarction, the absence of neutrophils is linked to the absence of resolution of inflammation, and to the ventricular remodeling and impaired cardiac output [40]. Among innate pro-inflammatory cells active in HF inflammation, some evidence has pointed towards the importance of eosinophils: indeed, a worse outcome after MI has been reported in patients with an increased eosinophil count [41]; moreover, eosinophils have been detected in intra-coronary thrombosis [42]. The evidence is still anecdotal, and further studies are necessary to clarify the effective function of eosinophils in mediating myocardial inflammation and injury. Finally, among innate immunity mediators, the role of TLRs is being progressively studied and understood more [43]. TLR2 is expressed in cardiomyocytes and in vascular epithelial cells, it participates in oxidative stress and represents a strong contributor in the development of HF and in cardiac remodeling after myocardial infarction [44]. TLR3 is involved in the protection of the heart against viruses, but it may also contribute to myocardial inflammatory damage [45]. Finally, TLR4 is the best studied TLR in heart diseases. It is activated by LPS and it is involved in myocardial inflammation, MI, HF, and ischemia/reperfusion injury [46]; furthermore, TLR4 is increased in advanced stages of HF [47].

Not only the innate, as discussed above, but also adaptive immunity has shown to have a role in the starting, maintaining and/or reducing inflammatory events during CV diseases.

A subset of T helper cells, known as Th17 cells, shows pro-inflammatory functions via IL-17 production; during ischemia, in the cardiac microenvironment, there is an increased production of pro-inflammatory cytokines, such as IL-1β, IL-6 and TNF-α, that promote Th17 cells maturation [48]. 

Recently, data from the literature have shown that higher levels of IL-17A are present in HF, especially in the higher classes of NYHA score [49]. IL-17A can stimulate fibroblasts to promote monocyte and macrophage migration to the myocardium, leading to ventricular remodeling. Moreover, the IL-17/IL-23 axis, well characterized as a pro-inflammatory pathway in several human pathologies [50], is able to further enhance ventricular remodeling, by recruiting neutrophils and macrophages, inducing the production of pro-inflammatory cytokines, cardiomyocyte apoptosis and fibrosis [51]. These modifications are dramatically more evident in dilated cardiomyopathy, a condition usually present in the late stages of HFrEF [52]. 

On the other hand, IL-17 could play a role not only in HFrEF but also in HFpEF. In a recent study conducted by Xu et al. [53], in a total of 120 patients with preserved ejection fraction, the combination of IL-17 and IL-6 levels correlated with the probability of left ventricular diastolic dysfunction (LVDD) [54]. Higher levels of these cytokines also showed a correlation with higher levels of fibrotic markers in this subset of patients, such as MMP-9, procollagen type I and type III [53].

Adaptive immunity can also play a protective role, and NK cells are among the cells that have a similar action. There is evidence that NK cells in HF patients are able to limit eosinophil recruitment to the inflamed myocardium, inducing apoptosis or altering their chemokine production [54].

DCs may also have a positive role in HF. In fact, these cells contribute to the immune response to specific pathogens and maintain self-tolerance. In particular, DCs may play a protective role in post-infarction inflammation recruiting neutrophils and monocytes at the inflammatory site [55]. Additionally, depletion in the number of DCs is associated with a higher mortality and left ventricular remodeling, heart healing, and increased MMP-2 and MMP-9 [51].

Tregs are able to suppress auto-reactive immunity and obtain the immune tolerance against self-antigens and non-self-antigens. Circulating Tregs are reduced in patients with HF and the number of Tregs inversely correlates with the severity of HF and does not depend on HF etiology [56]. Yet, in HFrEF, a lower number of Tregs are related to a worsening of HF [48]. It also has been demonstrated that Tregs are able to diminish ventricular remodeling in mice after MI [57] and reduce HF progression and right ventricular hypertrophy [58]. 

Particularly in the case of acute injury [59], cardiomyocytes are able to produce high level of TGF-β, which induces fibroblast activation, collagen production and extracellular matrix deposition [60], mechanisms that all lead to cardiac fibrosis. The high levels of TGF-β and IL-10 seen at site of injured myocardial tissue is counterproductive in terms of Treg functionality: when Tregs are chronically stimulated by cytokines, but cannot activate (in this case, injured cardiomyocytes limit the action of the cytokines), they lose their immunosuppressive ability. Furthermore, Liao et al. [52] have demonstrated that the severity of HF is related to a minor amount of circulating Tregs and a higher expression of histone deacetylase (HDAC9) mRNA. HDAC9 has been linked to atherosclerosis and many other heart diseases [59,61]. Moreover, a reduced number of Tregs is demonstrated both in ischemic and non-ischemic HF and their levels are correlated with a better EF and lower levels of brain natriuretic peptide (BNP). On the other hand, HDAC9 levels are related to a worse EF, to higher levels of BNP [52], and to higher classes of the NYHA score.

However, the role of Tregs in cardiac injury remains controversial. A small size study performed by Gorzin et al. [62] has shown no differences in the expression of Treg marker FoxP3 between HF and healthy subjects and did not find any correlation of its expression with HF severity. Levels of FoxP3 were, instead, significantly lower in ischemic HF than in not ischemic ones. The same authors did not observe any significant differences in the transcription factor retinoic acid receptor-related orphan receptor-gammat (ROR-gammat) that is a marker of Th17 cells, between HF and healthy subjects [62]. In a similar way, Zhu et al. have not observed significant differences in circulating levels of Th17 cells and IL-17 related cytokine and ROR-gammat in HF with respect to healthy subject [63]. Conversely, Li et al. showed that an imbalance between Tregs and Th17 cells exists both in patients with HFpEF and with HFrEF [46]. In fact, as above described, on the one hand, circulating Th17 cells are significantly increased in patients with HF, and on the other hand, Tregs significantly decrease in the same subjects. Thus, cardiac inflammation is responsible for the alteration in the Tregs/Th17 cell ratio that is the trigger of cardiac remodeling [64].

Overall, it can be argued that the pathogenesis of the consequences of myocardial injury comprises an impairment in the activation of several immune response’s pathways, comprising both molecules derived from innate and adaptive immunity cells and lymphocytes, whose activity can worsen myocardial fibrosis and lead to ventricular dysfunction; on the other hand, certain molecules of immunity could exert a protective role against deposition of extracellular matrix and disable further immune activation, suggesting a potential therapeutic role. 

## 5. Perspectives of Immune Modulation in Heart Failure

In a review concerning the role of inflammation in patients with rheumatoid arthritis (RA) and heart failure, Chen et al. [65] report some studies which have demonstrated a protective role of methotrexate in RA patients undergoing treatment with it, with a lowering in cardiovascular risk and a lower risk of HFpEF (but not HFrEF) [65]. However, this protective effect has been proven only in patients with RA and could be mediated by the benefits given on the rheumatologic condition and due to a higher inflammatory activity in these patients [65].

In the case of biological drugs such as the monoclonal cytokine inhibitors, Hanna and Fragogiannis [66] have suggested that a promising role could be played by the inhibitors of IL-1 and IL-6, canakinumab and tocilizumab, while the observations acquired in humans failed to prove benefits from treatments with TNF-alpha inhibitors in heart failure [66]. 

Moreover, pirfenidone, an oral antifibrotic agent which inhibits the proliferative effects of TGF-beta and has shown favorable effects on myocardial fibrosis in preclinical models, has been recently studied in a phase II clinical trial in patients with HFpEF [67]. In this phase II study, pirfenidone showed favorable effects on myocardial fibrosis and levels of NT-proBNP. However, these data need to be confirmed by further studies [67].

Overall, most of the evidence concerning the newest biological inhibitors derives from experiments in murine models and the clinical translation of these observations to human models is difficult and challenging, due to the heterogeneity of the heart failure, the lack of a clinical subset of patients with a specific pathway of cardiac inflammation (which could probably benefit from specific molecular inhibitors), and the need for expensive and complex clinical trials to state the effective benefits [66].

## 6. Gut Microbiota Composition and HF

CV health is highly influenced by gastrointestinal functionality and vice versa. For example, constipation has been associated with an increased CV risk and in particular to the development of high blood pressure (BP) [68], while HF may be responsible for alterations in intestinal function, bowel wall edema, and intestinal dysbiosis [69]. Chronic liver disease has also been linked to CV dysfunction and HF; the mechanism seems to involve both liver X receptors (LXR) and gut microbiota alterations [70].

The crosstalk between the gut microbiota and the CV system is complex: both an obesogenic diet and malnutrition may promote HF, suggesting that an unhealthy diet and the consequent alterations in the composition of gut microbiota have a causative role in HF pathogenesis. Moreover, patients with chronic HF face major changes in the composition of gut microbiota, and this may be considered as the cause but also a consequence of HF [71]. As stated above, HF can determine systemic microcirculatory disturbances, also affecting the splanchnic circulation, which can promote bacterial translocation, enhancing systemic inflammation. In turn, inflammation can favor further microbial translocation, thus determining a pathologic vicious circle [72]. Overall, patients with chronic HF have an increased number of pathogenic bacteria, such as *Campylobacter* spp, *Salmonella* spp, *Shigella* spp, *Yersinia enterocolitica*, and *Candida* spp. These alterations were significantly linked to the severity of HF, according to the NYHA scale [73]. Moreover, the presence of *Chlamydia pneumoniae* has been linked to an increased risk for the CV system; however, treating it did not bring any consistent benefit to patients, demonstrating even more that the crosstalk between the CV system and the microbiota is not as straightforward as could first appear [74].

One of the mechanisms through which microbiota can promote the development of HF is arterial hypertension [75]. Animal models have shown that gut dysbiosis precedes arterial hypertension and that administration of short-chain fatty acids (SCFAs), such as acetate, propionate and butyrate, is able to improve blood pressure (BP) control, while a diet rich in salt determines a loss of the beneficial bacteria Lactobacilli spp [76]. Additionally, other tissues were directly impacted by the administration of SCFAs: the renin–angiotensin–aldosterone system (RAAS) and pro-inflammatory IL-1 signaling pathway in the kidney were down-regulated, as well as mitogen-activated protein kinase (MPK) and TGF-β signaling in the heart [77]. SCFAs may also directly regulate BP control, through interaction with the host G-protein-coupled receptors (GPCRs), olfactory receptor 51E2 (OR51E2, also known as OLFR78) and free fatty acid receptor 3 (FFAR3; also known as GPR41) [78,79]. SCFAs also appear to be involved in the modulation of the response to ischemic insults [78].

Bile acid production is also regulated by the gut microbiota. Additionally, it appears to play a causative role in the pathogenesis of different CV diseases, particularly HF [80]. One of the mechanisms through which bile acids act is through bile acid receptor (FXR) signaling [81], even though it is not clear whether the role of FXR is positive or not. An increased FXR signaling has been linked to a reduction in nuclear factor-κB (NF-κB), which is responsible both for inflammation and direct cardiac remodeling, particularly in BP-independent hypertrophy. While studies have reported a positive impact on HF outcomes when FXR-signaling is implemented [82], other authors report that, in vitro, NF-κB instead induces myocardial apoptosis [83]. Some FXR agonists are being tested in patients with steatohepatitis, and the results have been encouraging in terms of CV health, in particular on atherosclerosis [84]. Yet, the functions and roles of FXR are still not completely understood, so it still not yet clear if the positive actions of FXR outweigh its side-effects [85].

Furthermore, microbiota can be related to the pathogenesis of coronary artery disease; in fact, in the same patient, the same microbial species have been found in both the plaque and the gut microbiota, representing a possible relationship between gut microbiota and the pathogenesis of arterial diseases. In patients with atherosclerosis, the species found more abundantly than in normal individuals are represented by *Streptococcus*, *Collinsella*, *Veillonella*, and *Chryseomonas* [86].

Another mechanism through which microbiota can promote the development of HF is atrial fibrillation [87], which is associated with a high concentration of *Proteobacteria*, *Actinobacteria* and *Firmicutes*, *Streptococcus*, *Haemophilus*, *Alistipes*, *Enterococcus*, and *Klebsiella* [88]. Gram-negative lipopolysaccharides (LPS) are endotoxins able to trigger NLRP3 inflammasomes with consequent caspase-1 activation and the production of IL-1β and IL-18. Interleukins increase the gut permeability, which allows the passage of LPS into blood with consequent activation of TLR-4 and nuclear factor-kappa B (NF-κB). These actions cause vascular inflammation with myocyte apoptosis, fibrosis, and enlargement [89].

Cardio-renal syndrome (CRS) is a well-known disease in which chronic kidney disease (CKD) and HF interact with each other. Gut microbiota plays a complex role in this interaction. It has been well established that microbial urease transform urea in ammonia and ammonium hydroxide, which on the one hand, promote gut tight junction disruption and microbial translocation with the consequent increased inflammatory status, while on the other hand, worsen the kidney functionality [90]. Gut microbiota also produces non-dialyzable protein-bound uremic toxins, such as indoxyl sulfate and p-cresyl sulfate, which appear to have a causative role in the genesis of CRS [91]. Mechanisms through which the microbiota modulates CV health in general and HF in particular may act both in direct and indirect way [92].

An important gut-derived mediator influencing the CV system is trimethylamine oxide (TMAO). TMAO is a gut microbe-dependent metabolite of dietary choline and other trimethylamine-containing nutrients and provides an example of direct and indirect mediation: it plays an important role in CV health, as it has also been linked to many diseases, e.g., HF, hypertension, atherosclerosis, and dyslipidemia [93,94], and it seems to be involved both in the pathogenesis and in the outcomes of these diseases [95]. In patients with HF, high levels of TMAO are indeed linked both to the clinical course of the pathology itself and to its mortality [96]. The suggested mechanisms through which it may act are, among the direct ones, the enhancement of the expression of the scavenger receptor A (SR-A) and cluster of differentiation (CD)-26, the activation of caspase-1-activating NLRP3 inflammasome and the induction of T-cell differentiation [97]. TMAO has also been linked to platelet malfunction and hypercoagulability [98], and it has been studied in acute HF, in which it appears to act as a promoter of inflammation [99], acting prevalently as an indirect promoter of disease. TMAO could also be involved in the pathogenesis of CRS. In this way, Stubbs et al. observed that increased levels of TMAO have a significant direct correlation to progression of kidney dysfunction, worsening the overall condition of patients with CRS [100]. TMAO has definitely been linked to HF, both in its pathogenesis and its outcomes, even though the mechanisms are not yet fully understood. It is hypothesized that TMAO may play a direct role on tissue functionality, which is consistent with the finding that diastolic functionality is impaired in patients with high levels of circulating TMAO [101]. Even though the underlying mechanisms are not clear yet, the causative role of TMAO in HF has been proven in murine models; Organ et al. have demonstrated that a choline-rich diet and a TMAO-rich diet impact in a similar way the CV health of the animals in study, significantly worsening the outcomes of HF [57]. Overall, it is now well known that high levels of TMAO are directly correlated to severity of HF [102]. Interestingly, patients with HF have worse outcomes overall, even if the levels of TMAO are reduced and independently from the underlying etiology [101]. The correlation between higher levels of TMAO and a poorer prognosis of HF has recently been evidenced even by a meta-analysis performed in 2020 by Li et al. [103]. Interestingly, the risk of major cardiovascular events (MACEs) was still significantly higher in those with higher TMAO levels even after adjusting MACE risk for renal function.

TMAO has been linked to advanced stages of HF, while in the early stages of chronic HF the main findings have been increased levels of lipopolysaccharide (LPS) [104].

LPS primarily acts to indirectly promote inflammation. In fact, LPS binds to lipopolysaccharide-binding protein (LBP), which in turn can increase the levels of several pro-inflammatory proteins, in particular TNF-α [105]. LPS and other bacterial endotoxins are able to induce the production of numerous additional pro-inflammatory cytokines, such as IL-1, IL-6, and IL-12, which are in turn linked to an increase in the number of circulating CD14+ monocytes and soluble CD14+ receptor (sCD14), hallmarks of poor CV recovery [97]. LPS has been also linked to adverse CV events in patients who had already experienced a myocardial infarction (MI), and it can be considered both a marker of inflammation and bacterial translocation. Interestingly, MI patients do not necessarily show an increase in dangerous microbial species but rather an increase in the number of microbes normally present in the GI, such as *Lactobacillus*, *Bacteroides*, and *Streptococcus* spp [106]. Even though these alterations do not appear to hold a causative role, only being an epiphenomenon, they have been considered as markers of disease (particularly sCD14 and LPB), and they appear to be good indicators of CV health and metabolic status [107]. The metabolic status is, indeed, linked to an increase in CV diseases: an obesogenic diet, in particular, is an element which could drive the gut microbiota to promote acute HF. Karin et al. have demonstrated that rats who follow a high-calorie and high-fat diet face an alteration of the neutrophil to lymphocyte ratio in the gut, a reduction in the number of macrophages and a dysregulation of isoprostanoid, lipoxygenase, and several cytokines [108]. On the other hand, malnutrition has also been linked to HF and to microbiota alterations typically linked to HF [73,109].

Overall, patients with HF present with a different microbial population when compared to the general population: *Eubacterium rectale* and *Dorea longicatena* are less abundant in patients with HF and, in general, SCFA-producing bacteria are reduced. This appears consistent with the hypothesis that gut microbiota plays an important role in the pathogenesis of HF through inflammation.

HF patients present decreased levels of several gut microbial species [110] with a significant decrease in *Coriobacteriaceae*, *Erysipelotrichaceae*, and *Ruminococcaceae* spp, *Blautia*, and *Collinsella* [71]. Moreover, butyrate-producing bacteria, such as *Lachnospiraceae* spp, have been shown to be reduced [111]. Other decreased bacterial species in HF patients are *Oscillibacter* spp and *Sutterella wadsworthensis* [112].

Variations in gut microbiota are also involved in the transition between arterial hypertension and hypertensive heart failure. In a recent study performed by Gutierrez-Calabres et al. in murine models [113], the transition between compensated arterial hypertension and heart failure in rats was preceded by markers of gut dysbiosis such as the alteration of the *Firmicutes/Bacteroidetes* ratio and rise in markers of intestinal permeability, before cardiac manifestations of heart failure. According to this study, certain bacterial taxa could be identified as a marker of spontaneously hypertensive heart failure in hypertensive individuals [113].

However, the alterations of gut microbiota in HF also need to be examined in the contest of ageing. There is evidence that gut microbiota changes with age in patients with HF. Kamo et al. observed that younger patients with HF had a microbiota richer in *Faecalibacterium prausnitzii* and *Clostridium clostridioforme* than older patients, whilst *Lactobacillus* spp appears to be more present in the latter [74].

Analysis of gut microbiota in HF patients also needs to take into account not only the role of diet but also the role of drugs and inter-individual differences. Particularly, the relation between drugs and gut microbiota is bidirectional: on the one hand, drugs alter the microbiota, and on the other, microbiota can in turn alter the response of the organism to drugs. It has been shown, for example, that *Eggerthella lenta* is implied in patients’ response to digoxin [114], and a Dutch study pointed out that the metabolism of drugs used to control HF, such as angiotensin-converting enzyme inhibitors, β-blockers, and angiotensin II-receptor blockers, varied significantly with variations in microbiota composition [115].

## 7. Gut Dysbiosis, Inflammation and Cardiovascular Diseases

Gut microbiota (GM) play a pivotal role in balancing not only intestinal immunological response at gut surface but also the individual inflammatory status in the course of pathologies involving organs different from the gut or in systemic diseases [5]. On the other hand, when a pathologic or physiologic condition induces a variation in GM composition or an abnormally permeable gut mucosa (“leaky gut”), GM can exert an influence on pre-existing diseases or increase the risk of pathological conditions [5].

There is a strong crosslink between GM alterations and cardiovascular diseases, and gut dysbiosis represents a new interesting mechanism in the modulation of cardiovascular risk [116]. The paradigm of the interaction between GM alterations and cardiovascular diseases can be considered the inflammatory bowel diseases (IBDs), which are mainly represented by Crohn’s disease and ulcerative colitis [117]. Indeed, patients with IBD present a higher risk of atherosclerotic coronary and peripheral artery disease, arterial hypertension, deep venous thrombosis and arrhythmias, which can be all causes of heart failure [117], and all these conditions are considered extraintestinal manifestations of the disease.

IBDs are conditions associated with an altered gut permeability and an altered composition of GM due to gut inflammation, with a loss of microbial diversity [117,118].

An altered gut permeability can induce a translocation of bacterial species and microbial products through systemic circulation, resulting in systemic inflammation and possibly in alterations of glucose and lipid metabolism [116]; furthermore, a role in enhancing atherosclerotic process by microbial translocation has also been proposed [116]. Notably, in patients with coronary disease, GM species can be found in their atherosclerotic plaques and the guts of these patients show an impairment in certain GM species and therefore gut dysbiosis [116]. On the other hand, gut dysbiosis can be responsible for an impairment in the production of metabolites influencing atherosclerotic process and platelet aggregation [117,118]. The possible role of TMAOs in inducing atherosclerosis and platelet dysfunction has already been mentioned. Moreover, another recognized mechanism of the atherosclerotic process which can be influenced by gut dysbiosis is the production of neutrophil extracellular traps (NETs), which can suppress neutrophil activation and reduce the atherogenic process and are produced by GM [117]. GM also induce the production of Toll-like receptors 2 and 4, which can upregulate inflammatory processes and atherosclerosis [118]. Thus, the modulation of gut microbiota by gastrointestinal or systemic conditions affecting intestinal immunological niche could be the key to the modified cardiovascular risk of these patients. GM represents a new, interesting actor in the modulation of cardiovascular risk and cardiovascular diseases.

## 8. Gut Microbiota, Innate Immunity and HF

It is important to underline that a bidirectional interplay between gut dysbiosis and HF occurs: gut dysbiosis is able to worsen HF, but HF may be responsible for mucosal barrier alterations, due to the decreased cardiac output, intestinal hypoperfusion, hypoxia, and mucosal edema [92].

This model is characterized by two different sites of inflammation, the heart and the gut, constituting the ‘heart–gut axis’. Thus, heart injury and the consequent myocardial inflammation triggers and recruits innate immunity from peripheral circulation. Gut dysbiosis is also responsible for the systemic circulation of several pro-inflammatory cytokines and mediators of inflammation, such as IL-1, IL-2, IL-6, and TNF-α, and bacterial components, such as LPS, flagellin, peptidoglycans, etc. [12,119]. These bacterial components in turn trigger innate immunity by TLR and NOD receptor activation. TLR and NOD activation by DAMPs has shown to be associated with the development of insulin-resistance and hyperlipidemia, which are two mechanisms commonly involved in CV diseases [120].

At the site of inflammation, neutrophils seem to be the mostly recruited cells, and are responsible for ischemic damage via ROS production. Moreover, neutrophils may also recruit monocytes and macrophages at the site of heart inflammation, which in turn are associated with further production of pro-inflammatory cytokines [34].

The above-described picture is intricate and shows that the cardiac microenvironment may be considered as a complex system, in which several components act in mediating inflammation and repair, such as innate and adaptive immune cells, cytokines, mediators of inflammation, and bacterial components. Hence, we can consider the cardiac microenvironment as an ‘immunological niche’. Thus, the cardiac immunological niche is characterized by an intense cellular trafficking from systemic circulation and, cells that are normally considered pathological may become beneficial, while cells that are normally beneficial may become pathological, as consequence of the different activity of the same cytokines on the different immune cells recruited at the site of inflammation (Figure 1).

## 9. Conclusions

Gut microbiota can influence CV health both through direct and indirect mechanisms. The role of gut microbiota in shaping the immune system in general is well known, and it has also been studied in many diseases.

A bidirectional relationship between the gut microbiota and immune system exists, and innate cells regulated by the cytokine cocktail present at the injured cardiac tissue are responsible for cardiac remodeling that worsens HF. Hence, as consequence of the presence of a ‘heart–gut axis’, gut dysbiosis, due to the pathological condition of the leaky gut, determines the systemic circulation of bacterial components that are able to recruit pro-inflammatory immune cells and cytokines to cardiac tissue. However, immune cells are characterized by high plasticity for which each cell may exert different functions according to the local cytokine’s microenvironment, resembling Dr. Jekyll and Mr. Hyde. Thus, this complex interplay involving several immune cells, cytokines, mediators of inflammation, and gut microbiota characterizes the model of the heart immunological niche. Many studies have tried and are trying to evaluate whether microbiota manipulation can have a positive impact on CV health [121,122]. The PROICA study (NCT01500343) analyzed the use of probiotic *Saccharomyces boulardii* in patients with chronic HF. Patients administered probiotics had a significant reduction in cholesterol, uric acid, and creatinine levels as well as left atrial diameter, while the LVEF improved. These results are encouraging, as the treatment appeared to be safe and well tolerated [123]. The TIPTOP trial (NCT00469261) instead studied the role of doxycycline therapy in patients who had experienced an acute MI to prevent left ventricle remodeling; this study also found encouraging results as far as adverse left ventricle remodeling is concerned [124]. Other studies on the modulation of gut microbiota in HF have been carried out. In particular, the role of ursodeoxycholic acid (UDCA) in patients with chronic HF (NCT00285597) has been explored. UDCA appeared to be well tolerated and is able to improve peripheral perfusion in patients [125].

In summary, data demonstrate the strong relationship between environmental factors, such as diet, and gut microbiota composition in inducing or promoting CV health and disease. Thus, modifying the gut microbiota composition may result in the identification of a way by which it will be possible ensure CV health or even treat CV disease.

In conclusion, the intricate relationship in heart immunological niche may be considered as the basis of a new pathogenetic model for a modern concept of precision medicine to better understand CV pathologies and develop new horizons in the treatment of these diseases.

## Figures and Tables

**Figure 1 biomedicines-10-02712-f001:**
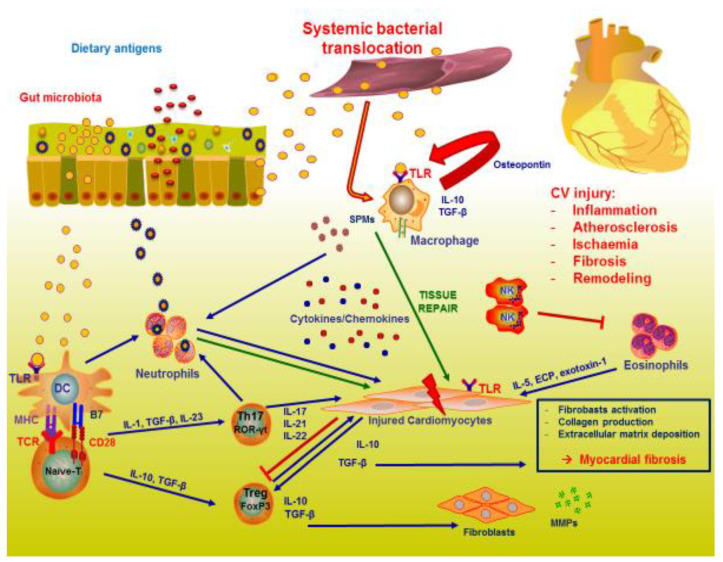
The complex interplay between gut microbiota and immunity in heart failure. A bidirectional interplay between gut microbiota and mucosal immunity occurs. In pathological condition, as consequence of bacterial translocation, innate and adaptive immune systems are activated, involving several cells, such as neutrophils, eosinophils, macrophages, dendritic cells, and T cells and their mediators. Cardiomyocytes are the final target of this complex activation of tissue inflammation. Injured cardiomyocytes are in turn responsible of further perpetuation of inflammatory process and innate cells and fibroblasts participate to tissue repair and cardiac remodeling.

**Table 1 biomedicines-10-02712-t001:** Interleukins and inflammation.

Cytokines		Effects	Productive Cells
Pro-Inflammatory Cytokines	TNF-α IL-1β	- Elevated levels of TNF-α are associated with HFrEF and its levels correlate with disease progression [14,17]; - Increase in inflammatory activity during myocardial injury [13,14,15]; - Stimulate Th17 cells differentiation and proliferation at tissue level, promoting myocardial injury; - Increase in myocardial stiffness and promotion of remodeling through fibrogenic factors and inhibition of MMP [13,14,15].	Neutrophils, Dendritic cells, Macrophages, Mononuclear cells
	IL-6 IL-8 IL-13 MIP-1α ICAM-1, VCAM-1 CCR2, CCL2 TLRs	- Elevated levels of these cytokines correlate with advanced stages of HF; - Promote an inflammatory status, recruiting pro-inflammatory cells; - Induce cardiac inflammation that can directly affect the hearth’s structure, increasing its stiffness (cardiac remodeling); - Negatively affect autophagic mechanisms in cardiac tissue; - Decrease the activity of the collagen degrading MMP-2 and MMP-9; - Increase the levels of free oxygen radicals; - Stimulate Th17 cells differentiation and proliferation at local tissue, promoting myocardial injury.	Neutrophils, Dendritic cells, Macrophages, Mononuclear cells
	IL-6	- Induce myocardial injury [13,14,15] through oxygen reactive products; - Its levels are associated with higher mortality for HF [23]; - Reduced levels after HF treatment [23].	Neutrophils, Dendritic cells, Macrophages, Mononuclear cells
Anti-Inflammatory Cytokines	IL-10	- Anti-inflammatory activity through down-regulation of pro-inflammatory mediators, inhibition of TNF-α, and of reactive oxygen products [17]; - Improve LV contractility and reduce the ischemic area [21]; - Elevated levels are associated with HF and advanced NYHA [19]; - Reduced levels after HF treatment [22].	Macrophages, Mononuclear cells, Tregs

## Data Availability

Not applicable.

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
