# Peer review of "The Role of Gut Microbiota in Heart Failure: When Friends Become Enemies"

_biomedicines, 2022, doi:10.3390/biomedicines10112712_

Round 1

Reviewer 1 Report

Cianci et al. in the article entitled "The role of gut microbiota in heart failure: when friends become enemies" presented the latest approach to the role of gut microbiota in the inflammatory process, which plays an important role in the pathogenesis and progression of heart failure. I recommend this article for publication.

Author Response

Many thanks for this comment.

Reviewer 2 Report

The impact of the gut microbiota on the human health is widely debated, and the major implications of its imbalance cover almost all systems and organs, from the nervous system to the circulatory system.

The topic is complexly approached and well managed, but it requires some improvements to the content, especially a better justification of the involvement of the intestinal microbiota in the modulation of cardiovascular diseases. In this context, I suggest a correlation between Gut dysbiosis, Inflammatory Bowel Diseases and heart failure (see the references suggested below).

·         Hao Wu, Tingzi Hu, Hong Hao, Michael A Hill, Canxia Xu, Zhenguo Liu, Inflammatory bowel disease and cardiovascular diseases: a concise review, European Heart Journal Open, Volume 2, Issue 1, January 2022, oeab029, https://doi.org/10.1093/ehjopen/oeab029

·         Masenga, S.K., Hamooya, B., Hangoma, J. et al. Recent advances in modulation of cardiovascular diseases by the gut microbiota. J Hum Hypertens (2022). https://doi.org/10.1038/s41371-022-00698-6

·         Bunu DM, Timofte CE, Ciocoiu M, Floria M, Tarniceriu CC, Barboi OB, Tanase DM. Cardiovascular Manifestations of Inflammatory Bowel Disease: Pathogenesis, Diagnosis, and Preventive Strategies. Gastroenterol Res Pract. 2019 Jan 13;2019:3012509. doi: 10.1155/2019/3012509. PMID: 30733802; PMCID: PMC6348818

I suggest to add, in Introduction, a schematic presentation of the main causes for Heart Failure.

Also, I suggest to add, in second chapter (HF and inflammation ) a table with the correlation between the pro and anti-inflammatory IL and  productive cells.

Also, I suggest an update of the bibliography, with more recent publications (2021-2022), for example:

·         Lu X, Liu J, Zhou B, Wang S, Liu Z, Mei F, Luo J, Cui Y. Microbial metabolites and heart failure: Friends or enemies? Front Microbiol. 2022 Aug 15;13:956516. doi: 10.3389/fmicb.2022.956516. PMID: 36046023; PMCID: PMC9420987.

·         Kazemian, N., Mahmoudi, M., Halperin, F. et al. Gut microbiota and cardiovascular disease: opportunities and challenges. Microbiome 8, 36 (2020). https://doi.org/10.1186/s40168-020-00821-0

Author Response

Reviewer 2:

The impact of the gut microbiota on the human health is widely debated, and the major implications of its imbalance cover almost all systems and organs, from the nervous system to the circulatory system.

  1. The topic is complexly approached and well managed, but it requires some improvements to the content, especially a better justification of the involvement of the intestinal microbiota in the modulation of cardiovascular diseases. In this context, I suggest a correlation between Gut dysbiosis, Inflammatory Bowel Diseases and heart failure (see the references suggested below).

  • Hao Wu, Tingzi Hu, Hong Hao, Michael A Hill, Canxia Xu, Zhenguo Liu, Inflammatory bowel disease and cardiovascular diseases: a concise review, European Heart Journal Open, Volume 2, Issue 1, January 2022, oeab029, https://doi.org/10.1093/ehjopen/oeab029
  • Masenga, S.K., Hamooya, B., Hangoma, J. et al. Recent advances in modulation of cardiovascular diseases by the gut microbiota. J Hum Hypertens (2022). https://doi.org/10.1038/s41371-022-00698-6
  • Bunu DM, Timofte CE, Ciocoiu M, Floria M, Tarniceriu CC, Barboi OB, Tanase DM. Cardiovascular Manifestations of Inflammatory Bowel Disease: Pathogenesis, Diagnosis, and Preventive Strategies. Gastroenterol Res Pract. 2019 Jan 13;2019:3012509. doi: 10.1155/2019/3012509. PMID: 30733802; PMCID: PMC6348818

  1. Thanks for this comment. We have added this paragraph:

7.Gut dysbiosis, inflammation and cardiovascular diseases

Gut microbiota (GM) play a pivotal role in balancing not only intestinal immunological response at gut surface, but also individual inflammatory status in the course of patholo-gies involving organs different from the gut or in systemic diseases [5]. On the other hand, when a pathologic or physiologic condition induces a variation in GM composition or an abnormally permeable gut mucosa (“leaky gut”), GM can exert an influence on pre-existing diseases or increase the risk of pathological conditions [5].

There is a strong crosslink between GM alterations and cardiovascular diseases, and gut dysbiosis represents a new interesting mechanism in modulation of cardiovascular risk [116]. The paradigm of the interaction between GM alterations and cardiovascular diseas-es can be considered the inflammatory bowel diseases (IBDs), which are mainly repre-sented by Crohn’s disease and ulcerative colitis [117]. Indeed, patients with IBD present a higher risk of atherosclerotic coronary and peripheral artery disease, arterial hypertension, deep venous thrombosis and arrhythmias, which can be all causes of heart failure [117] and all these conditions are considered extraintestinal manifestations of the disease.

IBDs are conditions associated with an altered gut permeability and an altered composi-tion of GM due to gut inflammation, with a loss of microbial diversity [117,118].

An altered gut permeability can induce a translocation of bacterial species and microbial products through systemic circulation, resulting in systemic inflammation and possibly in alterations of glucose and lipid metabolism [116]; furthermore, a role in enhancing ath-erosclerotic process by microbial translocation has also been proposed [116]. Notably, in patients with coronary disease, GM species can be found into their atherosclerotic plaques and the gut of these patients shows an impairment in certain GM species and therefore gut dysbiosis [116]. On the other hand, gut dysbiosis can be responsible for an impairment in the production of metabolites influencing atherosclerotic process and platelet aggrega-tion [117,118]. The possible role of TMAOs in inducing atherosclerosis and platelet dys-function has already been mentioned. Moreover, another recognized mechanism of the atherosclerotic process which can be influenced by gut dysbiosis is the production of neu-trophil extracellular traps (NETs), that can suppress neutrophil activation and reduce ath-erogenic process, and are produced by GM [117]. GM also induce the production of Toll-like receptors 2 and 4, that can upregulate inflammatory processes and atherosclerosis [118]. Thus, the modulation of gut microbiota by gastrointestinal or systemic conditions affecting intestinal immunological niche could be the key to the modified cardiovascular risk of these patients. GM represents a new, interesting actor in the modulation of cardio-vascular risk and cardiovascular diseases.

  1. I suggest to add, in Introduction, a schematic presentation of the main causes for Heart Failure.

Thanks for this comment. We have added these sentences:

The etiology of heart failure involves all the conditions that can induce an impair-ment of heart’s left or right ventricular filling, blood ejection or filling pressure [11]. These conditions include both diseases causing a direct myocardial injury or functional anoma-ly, such as acute or chronic ischemic cardiac disease, congenital or acquired cardiomyo-pathies, valve diseases, inflammatory heart diseases, infiltrative or accumulation disor-ders, cardiotoxicity due to chemotherapy or drug abuse or arrhythmias, and conditions which cause an indirect involvement of heart such as chronic pulmonary hypertension, anemia, endocrine and nutritional disorders, autoimmune diseases [8, 11}.

  1. Also, I suggest to add, in second chapter (HF and inflammation ) a table with the correlation between the pro and anti-inflammatory IL and productive cells.

We added a table, thank you for your suggestion

A summary of the effects of pro-inflammatory and anti-inflammatory cytokines in HF is reported in Table 1.

Table 1. Interleukins and inflammation

Cytokines

Effects

Productive cells

Pro-inflammatory cytokines

TNF-α

IL-1β

-          Elevated levels of TNF-α are associated to HFrEF and its levels correlate with disease progression [14,17];

-          Increase of inflammatory activity during myocardial injury [13-15];

-          Stimulate Th17 cells differentiation and proliferation at tissue level, promoting myocardial injury;

-          Increase of myocardial stiffness and promotion of remodeling through fibrogenic factors and inhibition of MMP [13-15].

Neutrophils, Dendritic cells, Macrophages, Mononuclear cells

IL-6

IL-8

IL-13

MIP-1α

ICAM-1, VCAM-1

CCR2, CCL2

TLRs

-          Elevated levels of these cytokines correlate with advanced stages of HF;

-          Promote an inflammatory status, recruiting pro-inflammatory cells;

-          Induce cardiac inflammation that can directly affect the hearth’s structure, increasing its stiffness (cardiac remodeling);

-          Negatively affect autophagic mechanisms in cardiac tissue;

-          Decrease the activity of the collagen degrading MMP-2 and MMP-9;

-          Increase the levels of free oxygen radicals;

-          Stimulate Th17 cells differentiation and proliferation at local tissue, promoting myocardial injury.

Neutrophils, Dendritic cells, Macrophages, Mononuclear cells

IL-6

-          Induce myocardial injury [13-15] through oxygen reactive products;

-          Its levels are associated with higher mortality for HF [23];

-          Reduced levels after HF treatment [23].

Neutrophils, Dendritic cells, Macrophages, Mononuclear cells

Anti-inflammatory

cytokines

IL-10

-          Anti-inflammatory activity through down-regulation of pro-inflammatory mediators, inhibition of TNF-α, and of reactive oxygen products [17];

-          Improve LV contractility and reduce the ischemic area [21];

-          Elevated levels are associated with HF and advanced NYHA [19];

-          Reduced levels after HF treatment [22].

Macrophages, Mononuclear cells, Tregs

  1. Also, I suggest an update of the bibliography, with more recent publications (2021-2022), for example:
  • Lu X, Liu J, Zhou B, Wang S, Liu Z, Mei F, Luo J, Cui Y. Microbial metabolites and heart failure: Friends or enemies? Front Microbiol. 2022 Aug 15;13:956516. doi: 10.3389/fmicb.2022.956516. PMID: 36046023; PMCID: PMC9420987.
  • Kazemian, N., Mahmoudi, M., Halperin, F. et al. Gut microbiota and cardiovascular disease: opportunities and challenges. Microbiome 8, 36 (2020). https://doi.org/10.1186/s40168-020-00821-0

We have added the suggested publications and the following ones:

Drapkina OM, Yafarova AA, Kaburova AN, Kiselev AR. Targeting Gut Microbiota as a Novel Strategy for Prevention and Treatment of Hypertension, Atrial Fibrillation and Heart Failure: Current Knowledge and Future Perspectives. Biomedicines. 2022 Aug 19;10(8):2019. doi: 10.3390/biomedicines10082019.

Zhao P, Zhao S, Tian J, Liu X. Significance of Gut Microbiota and Short-Chain Fatty Acids in Heart Failure. Nutrients. 2022 Sep 11;14(18):3758. doi: 10.3390/nu14183758.

Lu D, Zou X, Zhang H. The Relationship Between Atrial Fibrillation and Intestinal Flora With Its Metabolites. Front Cardiovasc Med. 2022 Jul 1;9:948755. doi: 10.3389/fcvm.2022.948755. eCollection 2022.

Reviewer 3 Report

The review paper summarizes the connection between immunity and heart failure, paying particular attention to the function of the microbiome in this clinical condition. The paper is well organized and written, however, there are a few suggestions:

Minor concerns:

1.     Line no 326 “One of the mechanisms through which microbiota…….” can author list a few types of mechanisms and examples

2.     Authors should also include a few studies related to the microbial metabolites associated with heart failure

Author Response

Reviewer 3:

The review paper summarizes the connection between immunity and heart failure, paying particular attention to the function of the microbiome in this clinical condition. The paper is well organized and written, however, there are a few suggestions:

Minor concerns:

  1. Line no 326 “One of the mechanisms through which microbiota…….” can author list a few types of mechanisms and examples.

We have added these sentences:

Furthermore, microbiota can be related to the pathogenesis of coronary artery disease; in fact, in the same patient, the same microbial species have been found in both the plaque and the gut microbiota representing a possible relationship between gut microbiota and the pathogenesis of artery duseases. In atherosclerotic patients, the species found more abundant than in normal populations are represented by Streptococcus, Collinsella, Veil-lonella, and Chryseomonas [Tanishq Kumar, Rajoshee R Dutta, Vivek R Velagala, Be-numadhab Ghosh, Abhay Mudey. Analyzing the Complicated Connection Between Intes-tinal Microbiota and Cardiovascular Diseases. Cureus. 2022 Aug 19;14(8):e28165. doi: 10.7759/cureus.28165. eCollection 2022 Aug].

Another mechanism through which microbiota can promote the development of HF is atrial fibrillation [Lu D, Zou X, Zhang H. The Relationship Between Atrial Fibrillation and Intestinal Flora With Its Metabolites. Front Cardiovasc Med. 2022 Jul 1;9:948755. doi: 10.3389/fcvm.2022.948755. eCollection 2022].that is linked to high concentration of Prote-obacteria, Actinobacteria and Firmicutes, Streptococcus, Haemophilus, Alistipes, Entero-coccus, and Klebsiella [Gut microbiota and metabolites in atrial fibrillation patients and their changes after catheter ablation. Huang K, Wang Y, Bai Y, et al. Microbiol Spectr. 2022;10:0]. Gram-negative lipopolysaccharides (LPS) are endotoxins able to trigger NLRP3 inflammasomes with consequent caspase-1 activation and production of IL-1β and IL-18. Interleukins increase gut permeability that allows the passage of LPS into blood with con-sequent activation of TLR-4 and nuclear factor-kappa B (NF-κB). These actions cause vas-cular inflammation with myocyte apoptosis, fibrosis, and enlargement [Zhao Y, Wang Z: Gut microbiome and cardiovascular disease . Curr Opin Cardiol. 2020, 35:207-18.10.1097/HCO.0000000000000720].

  1. Authors should also include a few studies related to the microbial metabolites associated with heart failure

Thank you for your comment. We have added some references concerning the role of metabolites in the paragraph “Gut microbiota composition and HF”, in the section in which we discuss the role of TMAO and LPS. In particular we have added:

Lu D, Zou X, Zhang H. The Relationship Between Atrial Fibrillation and Intestinal Flora With Its Metabolites. Front Cardiovasc Med. 2022 Jul 1;9:948755. doi: 10.3389/fcvm.2022.948755. eCollection 2022.

Zhao, P.,; Zhao, S.;, Tian, J.;, Liu, X. Significance of Gut Microbiota and Short-Chain Fatty Acids in Heart Failure. Nutrients. 2022 Sep 11;14(18):3758. doi: 10.3390/nu14183758

Huang, K.; Wang, Y,; Bai, Y. Gut microbiota and metabolites in atrial fibrillation patients and their changes after catheter ab-lation. Huang K, Wang Y, Bai Y, et al. Microbiol Spectr. 2022;10:0

Lu, X.;, Liu, J.;, Zhou, B.;, Wang, S., Liu Z, Mei F, Luo J, Cui Y. Microbial metabolites and heart failure: Friends or enemies? Front Microbiol. 2022 Aug 15;13:956516. doi: 10.3389/fmicb.2022.956516. PMID: 36046023; PMCID: PMC9420987.